# Gut Microbiota and Associated Mucosal Immune Response in Eosinophilic Granulomatosis with Polyangiitis (EGPA)

**DOI:** 10.3390/biomedicines10061227

**Published:** 2022-05-24

**Authors:** Elena Niccolai, Alessandra Bettiol, Simone Baldi, Elena Silvestri, Leandro Di Gloria, Federica Bello, Giulia Nannini, Federica Ricci, Maria Nicastro, Matteo Ramazzotti, Augusto Vaglio, Gianluca Bartolucci, Giacomo Emmi, Amedeo Amedei, Domenico Prisco

**Affiliations:** 1Department of Experimental and Clinical Medicine, University of Florence, 50134 Florence, Italy; elena.niccolai@unifi.it (E.N.); alessandra.bettiol@unifi.it (A.B.); simone.baldi@unifi.it (S.B.); elena.silvestri@unifi.it (E.S.); federica.bello@unifi.it (F.B.); giulia.nannini@unifi.it (G.N.); domenico.prisco@unifi.it (D.P.); 2Internal Interdisciplinary Medicine Unit, Careggi University Hospital, 50134 Florence, Italy; 3Department of Biomedical, Experimental and Clinical Sciences “Mario Serio” University of Florence, 50134 Florence, Italy; leandro.di.gloria@gmail.com (L.D.G.); matteo.ramazzotti@unifi.it (M.R.); augusto.vaglio@unifi.it (A.V.); 4Core Research Laboratory, Institute for Cancer Research and Prevention (ISPRO), 50139 Florence, Italy; federica.ricci@unifi.it; 5Department of Medicine and Surgery, University of Parma and Unit of Occupational Medicine and Industrial Toxicology, University Hospital of Parma, 43121 Parma, Italy; maria.nicastro@unipr.it; 6Nephrology Unit, Meyer Children’s Hospital, 50139 Florence, Italy; 7Department of Neurosciences, Psychology, Drug Research and Child Health Section of Pharmaceutical and Nutraceutical Sciences, University of Florence, 50139 Florence, Italy; gianluca.bartolucci@unifi.it

**Keywords:** eosinophilic granulomatosis with polyangiitis, ANCA-associated vasculitis, microbiota, T lymphocytes, short-chain fatty acids

## Abstract

Eosinophilic granulomatosis with polyangiitis (EGPA) is an anti-neutrophil cytoplasmic antibody (ANCA)-associated vasculitis. A genome-wide association study showed a correlation between ANCA-negative EGPA and variants of genes encoding proteins with intestinal barrier functions, suggesting that modifications of the mucosal layer and consequent gut dysbiosis might be involved in EGPA pathogenesis. Here, we characterized the gut microbiota (GM) composition and the intestinal immune response in a cohort of EGPA patients. Faeces from 29 patients and 9 unrelated healthy cohabitants were collected, and GM and derived metabolites’ composition were compared. Seven intestinal biopsies from EGPA patients with gastrointestinal manifestations were analysed to assess the T-cell distribution and its correlation with GM and EGPA clinical and laboratory features. No significant differences in GM composition, nor in the total amount of faecal metabolites, emerged between patients and controls. Nevertheless, differences in bacterial taxa abundances and compositional GM-derived metabolites profile were observed. Notably, an enrichment of potential pathobionts (Enterobacteriacee and Streptococcaceae) was found in EGPA, particularly in patients with active disease, while lower levels were found in patients on immunosuppression, compared with non-immunosuppressed ones. Significantly lower amounts of hexanoic acid were found in patients, compared to controls. The analysis of the immune response in the gut mucosa revealed a high frequency of IFN-γ/IL-17-producing T lymphocytes, and a positive correlation between EGPA disease activity and intestinal T-cell levels. Our data suggest that an enrichment in potential intestinal pathobionts might drive an imbalanced inflammatory response in EGPA.

## 1. Introduction

Eosinophilic granulomatosis with polyangiitis (EGPA), formerly Churg–Strauss syndrome, is a rare anti-neutrophil cytoplasmic antibody (ANCA)-associated vasculitis (AAV) affecting small- and medium-sized vessels, almost invariantly characterized by severe asthma and ear–nose–throat (ENT) involvement, associated with blood and tissue eosinophilia and systemic vasculitic manifestations [1].

Despite being classified among AAVs, only 30–40% of EGPA patients test positive for ANCA [2]. ANCA-positive and ANCA-negative patients have distinct phenotypes [3], reflecting a dual nature of EGPA [4]. Though not fully understood, EGPA pathogenesis seems to be the result of a complex interaction between genetic and environmental factors [1,5,6]. Recently, a genome-wide association study (GWAS) showed an HLA-DQ association in ANCA-positive EGPA, similar to other eosinophilic conditions. Conversely, in ANCA-negative EGPA, variants of genes encoding proteins with intestinal barrier functions have been described, suggesting that modifications of the mucosal layer and the consequent gut dysbiosis might be involved in disease pathogenesis [7].

The gut microbiota (GM) is an active component of the host immune system, being essential for immune education and homeostasis [8,9]. Particularly, GM metabolites—notably butyrate, propionate, and acetate—are the downstream mediators of the GM anti-inflammatory activity, with a pivotal role in the immunological cross-talk with the host [10]. For these reasons, dysbiosis, i.e., the imbalance of microbial communities, could promote dysregulated host immune responses, eventually leading to the development of immune-mediated diseases [11,12,13,14]. Accordingly, growing literature describes a role of microbiota dysbiosis in other systemic vasculitides [15,16,17]. Regarding AAVs, a dysbiotic nasal microbiota has been described, particularly in granulomatosis with polyangiitis [18,19,20,21,22], while data on the GM in AAVs, and particularly on EGPA, are missing.

On these bases, our study aimed to characterize the GM composition and the association between mucosal microbiota and tissue-infiltrating T cells in mucosal biopsies from EGPA patients.

## 2. Materials and Methods

### 2.1. Study Population and Setting

Adult patients meeting the American College of Rheumatology classification criteria for EGPA [23] or the criteria proposed in the MIRRA trial [24] were recruited between 2018 and 2020, at the Vasculitis Centre of the Interdisciplinary Internal Medicine unit of the Careggi University Hospital, Florence, Italy. Healthy controls (HC), selected among adult patients’ unrelated cohabitants and without chronic diseases, were also enrolled. Exclusion criteria included the presence of other systemic inflammatory disorders, active or recent infections or malignancies, and antibiotic and probiotic/prebiotic use in the previous 2 months. This study and its subsequent amendments were approved by the local Ethic Committee Comitato Etico di Area Vasta Centro (BEMIT study; Rif. Em.: 2017-360), and written informed consent was obtained from all the study subjects.

### 2.2. Clinical Assessment

At time of enrolment, all patients underwent a review of their medical history, a comprehensive laboratory work-up, and a standardized clinical evaluation. Disease activity was assessed using the Birmingham Vasculitis Activity Score (BVAS) [25]. Major organ involvement was defined as the presence of renal, neurological, cardiac, or gastrointestinal (GI) manifestations. Furthermore, ongoing pharmacological treatments were recorded, with particular reference to corticosteroids and traditional and biologic disease-modifying anti-rheumatic drugs (DMARDs).

### 2.3. Microbiota Characterization

At enrolment, all participants provided a faecal specimen. Mucosal biopsies obtained from routine colonoscopy in patients with GI symptoms were also obtained. Faeces and biopsies were immediately stored at −80 °C, until processing, and were shipped to the Wellmicro technology services (Bologna, Italy) that performed the total genomic DNA extraction (using the QIAamp PowerFaecal DNA kit (Qiagen, Hilden, Germany)), quantification and quality control (QIAdvanced system (Qiagen, Hilden, Germany)), generation of amplicons of the variable V3–V4 region of the bacterial 16s rRNA gene, and sequencing in paired-end (2 × 300) cycles on the Illumina MiSeq platform. Raw sequences were processed using QIIME2 2021.4. Briefly, sequencing primers were removed using the Cutadapt tool, and the DADA2 tool was used to perform the paired-end reads merging, filtering, and chimera-removal steps, after trimming nucleotides from forward and reverse reads based on the quality profiles. Hence, amplicon sequence variants (ASVs) were generated, and the V-search tool was used for taxonomic assignment using the SILVA database (release 138) as reference, with 0.99 identity threshold.

### 2.4. Short- and Medium-Chain Fatty Acids Analysis

The evaluation of faecal short- and medium-chain fatty acids (SCFA and MCFA, respectively) and the standard curves’ preparation was performed by an Agilent GC-MS system composed of a 5971 single quadrupole mass spectrometer, 5890 gas chromatograph and 7673 autosampler, through our previously described GC-MS method [26].

### 2.5. Generation and Characterization of T-Cell Clones from Intestinal Biopsies

Biopsies were dissociated with the Tumor Dissociation Kit, human (Miltenyi Biotech, UK), in combination with the gentleMACS™ Octo Dissociator (Miltenyi Biotech, Bergisch Gladbach, Germany), to obtain a gentle and rapid generation of single-cell suspensions. Tissue-infiltrating lymphocytes (TILs) were magnetically isolated with anti-human CD3 microbeads (Miltenyi Biotech, Bisley, UK) and cloned under limiting dilution [27]. The phenotypic (surface markers expression) and functional (cytokine production) characterization of isolated T-cell clones (Tcc) was performed by plychromatic flow cytometry, as previously described [28].

### 2.6. Statistical Analysis

Categorical variables were presented as absolute frequencies and percentages, and were compared between EGPA patients and HCs using the χ2 test for unpaired data. Continuous variables were presented as median value and interquartile range (calculated as the difference between the 75th and 25th percentiles of the data), and were compared using the non-parametric Mann–Whitney test. Statistical analyses of the bacterial communities were performed in R 4.1 (R Core Team, 2014) with the help of the packages phyloseq 1.36.0 [29], DESeq2 1.32.0 [30] and vegan 2.5-7 [31]. Packages ggplot2 3.3.5 [32,33] and pca3d 0.10.2 [34] were used to plot data and results. In rarefaction analysis, samples with a final slope with an increment in ASV number per reads <1 × 10^−5^ are arbitrarily defined as saturated. Cluster analysis and PCoA were performed on square-root-transformed percent abundance of ASVs of each sample. Differences in alpha diversity indices were tested through Wilcoxon–Mann–Whitney test. Bray–Curtis distance was used on normalized data (see above) to perform a PERMANOVA analysis at each taxonomic rank. The differential abundance analysis was performed with DESeq2 on raw ASV counts. Spearman correlation coefficients were computed to assess association between variables. *p*-values were corrected for multiple comparisons using the Benjamini–Hochberg FDR procedure for all analyses except for exploratory analyses performed within EGPA subgroups of interest [35].

### 2.7. Patient and Public Involvement Statement

This research was carried out without patient involvement. Patients were not invited to comment on the study design and were not consulted to develop patient-relevant outcomes or interpret the results. Patients were not invited to contribute to the writing or editing of this document for readability or accuracy.

## 3. Results

### 3.1. Study Population

Twenty-nine EGPA patients and nine sex/age-matched HC were enrolled. Their demographic and clinical characteristics are summarized in Table 1. The median age at enrolment was of 58 (IQR 35–90) years for EGPA patients and 62 (45–75) years for controls, and 55% and 44% of subjects in the two groups were female. Of note, 34% of EGPA patients were ANCA-positive. Most patients had a history of asthma and ENT involvement, with 83% also having major organ involvement. GI symptoms were present in seven patients, who had undergone routine colonoscopy. At the time of enrolment, 76% of patients presented an active vasculitis, defined by a BVAS >0. Moreover, all patients displayed blood eosinophilia, with 35% of patients displaying eosinophilia >500 cell/mm^3^. As for ongoing pharmacological treatments, 96% of patients were receiving oral corticosteroids, while 76% were (co)treated with DMARDs, mostly traditional.

### 3.2. Faecal Microbiota Composition

Our sequencing efforts in assessing microbiota composition encompassed a total of 2,583,630.00 reads for 38 faecal samples. After the pre-processing steps, 1,958,593.00 reads (75.8%) were available for further analysis. Rarefaction curves for observed ASVs (Appendix A) revealed that specimens were sufficiently sampled.

We first evaluated if differences existed in the faecal microbiota structure of EGPA patients, compared to controls. Regarding alpha diversity, estimated by Chao, Shannon, and the evenness indexes, we found similar richness and taxa distribution in the stool samples of patients and controls (Figure 1A).

Similarly, beta diversity analysis, assessed through the Bray–Curtis metric, and hierarchical clustering, did not reveal a marked sample separation between EGPA patients and HC (Figure 1B,C).

Taxonomic composition analysis revealed that five phyla prevailed in all samples from both EGPA patients and HCs (99% sequences), namely Firmicutes (71.8%), Bacteroidota (15.8%), Actinobacteriota (9.1%) Proteobacteria (1.2%), and Verrucomicrobiota (0.8%) (Figure 2A). No differences were observed at the phylum and class level between EGPA patients and HCs. Accordingly, the Firmicutes-to-Bacteroidota ratio (F/B) was not significantly different in patients, compared to controls (p = 0.321) (Figure 2B).

At lower taxonomic levels, Lachnospiraceae, Ruminococcaceae, and Bacteroidaceae were the most represented bacterial families in both groups, accounting in aggregate for a relative abundance of about 60%. Notably, EGPA patients displayed higher abundance of the bacterial orders Enterobacterales (p_adj_ = 0.001) and Lactobacillales (p_adj_ = 0.002), and an enrichment in the bacterial families Enterobacteriaceae (p_adj_ = 0.003), Lactobacillaceae (p_adj_ = 0.007), and Streptococcaceae (p_adj_ = 0.034). At genus level, *Enorma* (p_adj_ < 0.0001) and *Escherichia-Shigella* (p_adj_ = 0.016) were more abundant in EGPA patients compared to HCs, whereas *Enteroscipio* (p_adj_ < 0.0001) was significantly less represented (Figure 3A).

We then explored differences in faecal microbiota composition among subgroups of EGPA patients, stratified according to disease activity, ANCA status, presence of eosinophilia, current treatment with immunosuppressants, presence of GI symptoms, and major organ involvement.

Richness and taxa distribution were similar among the different patients’ subgroups (data not shown). Conversely, at lower taxonomic levels, we found that the Streptococcaceae family (p_adj_ = 0.006) was enriched in active EGPA compared to inactive patients. Moreover, patients without immunosuppressive treatment showed higher levels of the bacterial order Enterobacterales (p_adj_ = 0.005), whereas immunosuppressed patients had higher abundance of genera *Senegalimassilia* (p_adj_ < 0.0001) and *Holdemanella* (p_adj_ < 0.0001). Moreover, the latter was more represented in patients with a lower eosinophilic count (p_adj_ < 0.0001). Finally, in patients with major organ involvement, we found a significant increase in the genera *Peptococcus* (p_adj_ < 0.0001), *Desulfovibrio* (p_adj_ < 0.0001), *Gemella* (p_adj_ = 0.023), and *Lachnospiraceae_UCG-001* (p_adj_ = 0.031) (Figure 3B). No difference was observed between ANCA positive vs. negative patients (data not shown).

### 3.3. Microbiota-Derived Metabolites

Regarding GM-derived metabolites, the total amount of fatty acids was not significantly different in stools from EGPA patients, compared to stools from HC (42.61(27.92) vs. 51.05(30.45) μmol/g; p = 0.162). Assessing the percentage of each SCFA and MCFA, we found significantly lower amounts of hexanoic acid in EGPA (0.00(0.59) vs. 0.62(0.72) p = 0.015) in patients compared to HC, with no differences in the other considered fatty acids (Appendix A). Interestingly, levels of GM metabolites also differed among subgroups of EGPA patients. Specifically, patients with active disease showed higher amounts of valeric acid compared to those with inactive disease. As for disease manifestations, we found that patients with GI symptoms had higher levels of isobutyric and 2-methylbutyric acid compared to patients without GI symptoms. Moreover, patients with eosinophilia had significantly higher amounts of isobutyric, 2-methylbutyric, valeric and isovaleric acids. Conversely, no significant difference in SCFAs and MCFAs was found according to ANCA status, presence of major organ involvement, and immunosuppressive treatment (Appendix A).

### 3.4. Gut Microbiota and Mucosal T-Cell Response in EGPA Patients

In the subgroup of EGPA patients with GI manifestations, we assessed the mucosal microbiota and tissue-infiltrating T cells in mucosal biopsies. A total of 209,155.00 sequencing reads were obtained from seven mucosal samples, and 166,827.00 (79.8%) passed the pre-processing steps. The taxonomic composition revealed that the most abundant phyla were Firmicutes (49.8%), Bacteroidota (31.6%), Proteobacteria (15.6%), Actinobacteriota (1.6%), and Fusobacteriota (0.7%) (Figure 4A). At lower taxonomic levels, Lachnospiraceae, Bacteroidaceae, and Ruminococcaceae were the most represented bacterial families, collectively accounting for a relative abundance of about 68%, followed by Pseudomonadaceae and Enterobacteriaceae. Finally, the five most abundant genera were *Bacteroides* (26%), *Anaerostipes* (8%), *Pseudomonas* (7%), *[Ruminococcus]_gnavus_group* (6%), and *Escherichia-Shigella* (5%) (Figure 4B).

To evaluate the intestinal immune response, we expanded and cloned in vivo-activated TILs after tissue dissociation. We obtained a total of 314 Tcc. The median number of Tcc isolated from each patient was 28 ± 28.5. In total, 81% (255/314) of isolated Tcc were CD4+, namely T helper (Th), with a median number of Tcc of 24 ± 16, and 19% (59/314) were CD8+, with a median of 5 ± 11 (Figure 5A). Focusing on cytokine expression, 6% of CD4+ Tcc produced IL-17 alone (15/255, Th17), and 44% (111/255) produced IL-17 in combination with IFN-γ (Th1/Th17). Further, 26% (67/255) of CD4+ Tcc produced IFN-γ alone, 6% (15/255) produced IL-4 in combination with IFN-γ (Th0), and 2% (5/255) produced IL-4 alone (Th2). Sixteen percent (42/255) of CD4+ Tcc were Tregs. Regarding CD8+ Tcc, the majority was able to produce IFN-γ alone (40/59, 68%, Tc1), but a considerable percentage of them (32%) produced IL-17, alone (12%, 7/59) or in combination with IFN-γ (12/59, 20%, Tc1/Tc17) (Figure 5B).

Finally, the potential correlation between the abundance of patients’ clinical features, Tcc, and mucosal bacterial taxa was explored. A positive correlation was found between patients’ disease activity and the total number of Tcc (Rho = 0.901), and of Th0 (Rho = 0.898), isolated from mucosal samples (Figure 5C).

Regarding the correlation with mucosal GM (Figure 6), both the BVAS (Rho = 0.775) and the eosinophil counts (Rho = 0.821) positively correlated with the abundance of Bacteroidota. Moreover, the total Tcc number was positively related to the abundance of Bacteroidota (Rho = 0.964), Bacteroidaceae (Rho = 0.786), and *Bacteroides* (Rho = 0.786), and inversely with Lachnospiraceae (R = −0.821). Fusobacteriota showed a negative correlation with Th1 (Rho = −0.788) and a positive correlation with Treg (Rho = 0.867) and Tc1/Tc17 (Rho = 0.900). At lower taxonomic levels, the genus *[Ruminococcus]_gnavus_group* negatively correlated with Tc1 (Rho = −0.821) and Tc17 (Rho = −0.808), while the latter positively correlated with *Pseudomonas* abundance (Rho = 0.867) (Appendix A).

## 4. Discussion

Gut microbiota dysbiosis has been reported in systemic vasculitides, but no study has specifically focused on AAVs, or on EGPA, in particular [15].

ANCA-negative patients represent about 70% of EGPA patients; according to a recent GWAS, variants of genes encoding proteins with intestinal barrier functions have been described to be associated with ANCA-negative EGPA, suggesting a role for a disruption of the mucosal layer and the consequent gut dysbiosis in EGPA pathogenesis [7].

This study characterized the GM composition in EGPA patients compared to unrelated HC, and further investigated the association between mucosal GM and local immune response in EGPA patients with GI manifestations.

Our analysis of bacterial 16S in stools showed a similar GM profile in EGPA patients and controls, with no significant differences in terms of alpha or beta diversity. Accordingly, the taxonomic compositional analysis showed a comparable Firmicutes-to-Bacteroidota ratio (usually considered an indicator of dysbiosis) and phyla distribution, with a similar abundance of microbiota-derived metabolites.

Conversely, at a deeper taxonomic level, an enrichment of potential intestinal pathobionts emerged in EGPA patients. Namely, we found that EGPA patients had higher levels of Enterobacterales and Lactobacillales orders, specifically of the Enterobacteriaceae, Lactobacillaceae, and Streptococcaceae families. Evidence suggests that a dysregulation of the innate immune system can induce the overgrowth of Proteobacteria (such as Enterobacteriaceae), which can promote intestinal inflammation [36], as observed in inflammatory bowel disease [37,38,39]. Moreover, a higher abundance of Enterobacteriaceae has been reported in conditions characterized by a systemic inflammation, including in thoracic aorta aneurysm biopsies of patients with giant cell arteritis [40].

Here, we found that EGPA patients receiving systemic immunosuppressants had significantly lower levels of Enterobacteriaceae compared to non-immunosuppressed patients. These findings, while suggesting a potential association between these bacteria and the systemic inflammatory state, support also the hypothesis that an appropriate treatment of vasculitis could reshape the microbiota composition towards a healthier profile [41].

Another distinctive feature of EGPA was the enrichment in Streptococcaceae, particularly in subjects with active disease. Interestingly, this family includes *Streptococcus* spp., which is responsible for streptococcus-associated medium-vessel vasculitis [41], and that has been found increased in faecal samples of patients with active Kawasaki disease [42], a systemic small- to medium-sized-vessel vasculitis. Nevertheless, the specific mechanisms underlying the association between Streptococcaceae abundance and active EGPA remain unclear. At a deeper taxonomic level, we found that the genera *Enorma* and *Escherichia-Shigella* were enriched in patients compared to HC, whereas *Enteroscipio* was less represented. Interestingly, *Escherichia* and *Shigella* genera (which were assessed and referred to jointly since they have essentially the same 16S rRNA sequences [36]), are translocating facultative aerobic bacteria, and potentially pathobionts in susceptible subjects under specific environmental stimuli [43].

Notably, we also observed slight differences in the GM-derived metabolites’ profile in EGPA patients compared to controls, which according to some EGPA clinical and laboratory features, further suggest that the presence of a functional microbiota dysbiosis could have an impact on the immune state [44].

Additionally, we attempted to evaluate the potential association between mucosal GM and gut T-cell immune response in EGPA with GI involvement.

T cells play a key role in EGPA pathogenesis. In particular, EGPA is considered a Th2-mediated disorder, with high abundance of tissue-infiltrating Th2, and with CD4+ T cells able to produce, in vitro, elevated concentrations of IL-4, IL-5, and IL-13 [45,46]. In peripheral blood of EGPA patients, increased levels of Th2 and Th17 [47,48] paired with reduced levels of Tregs [49] are observed.

Here, we characterized for the first time the intestinal distribution of T-cell subsets in EGPA patients undergoing colonoscopy. Our results showed a high abundance of CD4+ T cells, the Th1/Th17 subset being the most represented. Overall, 16% of CD4+ T cells displayed a regulatory profile, and only 8% of clones produced IL-4. In a previous work, we extensively characterized the distribution of T cells in tumour tissue and adjacent mucosa of patients with colorectal cancer. Interestingly, the percentage of Tregs and IL-4-producing cells in EGPA patients was higher than in healthy mucosa, similarly to that detected in colorectal cancer [28]. However, the most interesting datum we observed was the high percentage of IL-17-producing cells at mucosal level in EGPA patients, compared to that of tumour and normal gut tissues, paralleling what had already been observed in other immune-mediated disorders [50]. Since infection and inflammation drive commensal-specific CD4+ T cells toward Th1 and Th17 differentiation [51], these data suggest that the Th1/Th17 axis, rather than a Th2 response, could drive inflammation in the intestinal mucosa of EGPA patients with GI manifestations [52,53]. Interestingly, T-cell clones significantly correlated with the abundance of some GM taxa, in particular Bacteroidota, which in turn correlated with disease activity. Conversely, Fusobacteriota exhibited a negative correlation with Tregs and Th1, suggesting that a low abundance of these bacteria could affect the immune balance. Some limitations should be considered when interpreting the results of this study. First, we used the sequencing of the highly conserved 16S bacterial gene, which has limited ability to determine species identity, and does not allow for the enumeration of functional content. Moreover, regarding our analysis on the mucosal immune response, a comparison with HC was lacking, since, for ethical reasons, no biopsy could be taken from healthy cohabitants. In addition, our data describe the association between microbiota and mucosal immune response, but no mechanistic link between variations in microbiota composition and disease-related immunological changes was investigated.

Despite these limitations, our study provides, for the first time, data on the GM composition in EGPA. Taken together, our results did not indicate a substantial intestinal taxonomic dysbiosis in EGPA patients compared to controls, though an enrichment of potential pathobionts sustaining inflammation was observed in EGPA, particularly in patients with active disease.

## 5. Conclusions

The analysis of the intestinal immune response revealed that a Th1/Th17-mediated response, rather than a Th2-driven inflammation, is more likely to drive the pathogenesis of EGPA-related GI manifestations. Although pathogenic mechanisms deserve further investigation, the association we found among intestinal T-cells levels, GM, and disease activity suggests that the microbiota composition could affect the intestinal immune response in EGPA or vice versa, paving the way for the identification of new elements in EGPA pathophysiology and the identification of novel therapeutic targets.

## Figures and Tables

**Figure 1 biomedicines-10-01227-f001:**
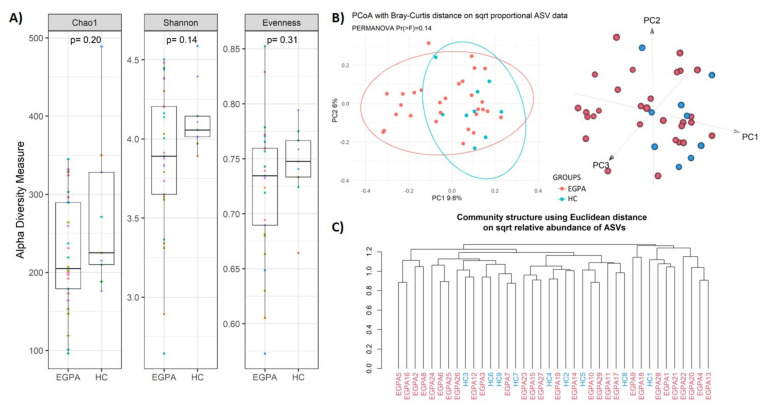
Alpha and Beta diversity analysis between EGPA patients and controls. (**A**) Boxplots showcasing alpha diversity indices (Chao1 index, Shannon index, Evenness) in fecal samples. Statistical differences were evaluated using paired Wilcoxon signed-rank test. *p*-values less than 0.05 were considered statistically significant. (**B**) Principal coordinates analysis (PCoA) according to the Bray–Curtis beta-diversity metric. Results of the permutational multivariate analysis of variance (PERMANOVA) are also shown based on the first two coordinates. Both 2D and 3D representation are provided. (**C**) Agglomerative cluster analysis using Euclidean distance as metric. EGPA = eosinophilic granulomatosis with polyangiitis; HC = healthy controls.

**Figure 2 biomedicines-10-01227-f002:**
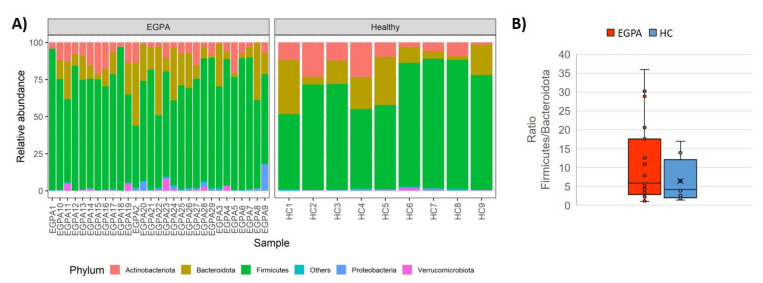
Phylum-level distribution in faecal samples. (**A**) Stacked bar plots displaying the average relative abundance of bacterial amplicon sequence variants (ASVs) identified at the phylum taxonomic level in EGPA patients and controls; (**B**) Box plot of Firmicutes-to-Bacteroidota ratios (median ± IQR). EGPA = eosinophilic granulomatosis with polyangiitis; HC = healthy controls; F/B = Firmicutes: Bacteroidota.

**Figure 3 biomedicines-10-01227-f003:**
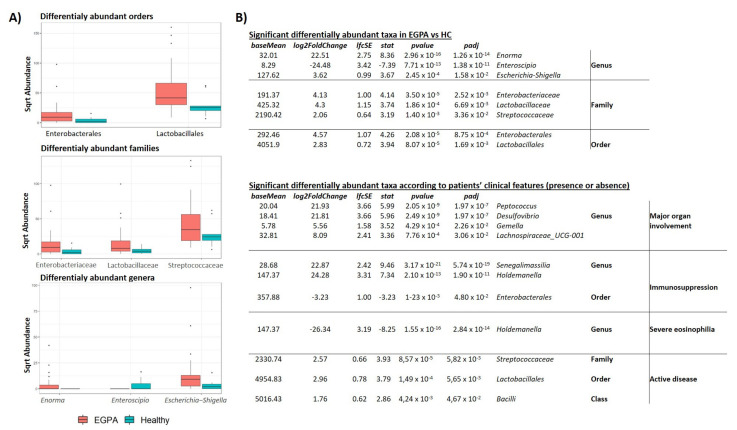
Significant differentially abundant taxa among EGPA patients and controls. (**A**) Boxplot showing the abundance of orders, families and genera associated with a statistically significant variation. (**B**) DESeq2 results of the differential abundance taxa in EGPA patients compared to controls and among patients with distinctive clinical features. EGPA = eosinophilic granulomatosis with polyangiitis; HC = healthy controls.

**Figure 4 biomedicines-10-01227-f004:**
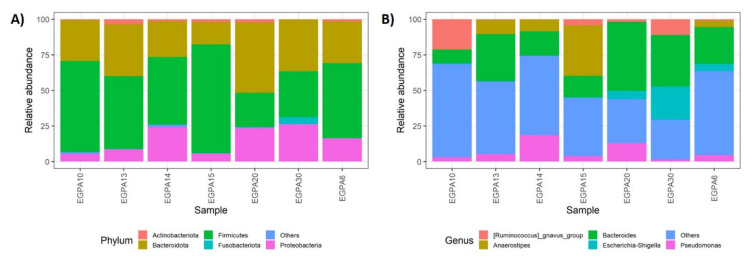
Stacked bar plots displaying the average relative abundance of bacterial amplicon sequence variants (ASVs) identified at the (**A**) phylum and (**B**) genera taxonomic level in the mucosal microbiota of EGPA patients. EGPA = eosinophilic granulomatosis with polyangiitis.

**Figure 5 biomedicines-10-01227-f005:**
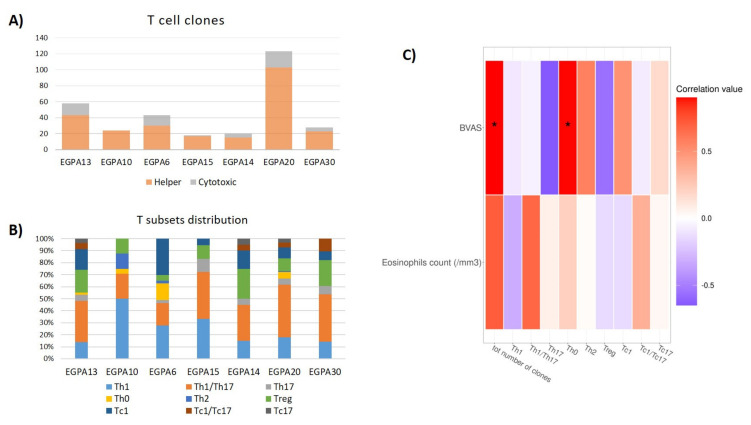
T-cell clones (Tcc) distribution in EGPA patients. (**A**) Barplot with the number of Helper and Cytotoxic Tcc isolated from each patient. (**B**) Barplot of T subsets distribution (percentage of total Tcc) in EGPA patients. (**C**) Heatmaps of correlations between patients’ Tcc and the BVAS and the eosinophil count (cell/mm^3^). EGPA = eosinophilic granulomatosis with polyangiitis; BVAS = Birmingham Vasculitis Activity Score. * *p*-value < 0.05.

**Figure 6 biomedicines-10-01227-f006:**
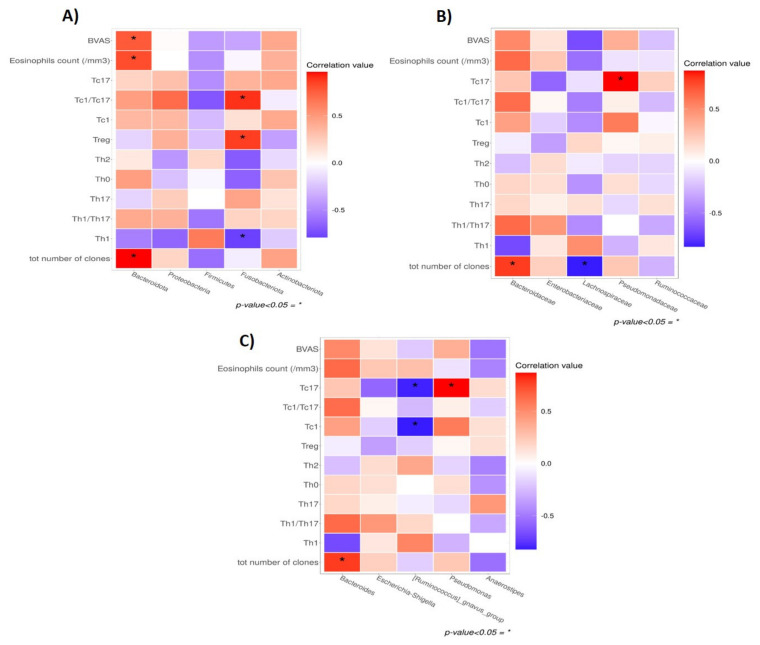
Correlation between bacterial taxa, T-cell clones (Tcc) subsets and clinical parameters. Heatmaps of correlations between the Tcc percentage and the relative abundance of the five most represented phyla (**A**) families (**B**) and genera (**C**). BVAS = Birmingham Vasculitis Activity Score. * *p*-value < 0.05.

**Table 1 biomedicines-10-01227-t001:** Clinical characteristics of enrolled EGPA patients and healthy controls. ANCA, antineutrophil cytoplasmic antibody; DMARDs, disease-modifying anti-rheumatic drugs; EGPA, eosinophilic granulomatosis with polyangiitis; ENT, ear–nose–throat.

Participants’ Characteristics at Time of Sampling	EGPA, n (% Out of 29)	Controls, n (% Out of 9)	*p*-Value
**Age** (years; median (IQR))	58 (55)	62 (30)	0.868
**Female sex**	16 (55%)	4 (44%)	0.980
**ANCA positivity**	10 (34%)	-	
**Disease manifestations in the medical history**			
Pulmonary	28 (97%)	-	
ENT	24 (83%)	-	
Neurological	18 (62%)	-	
General	17 (59%)	-	
Cutaneous	11 (38%)	-	
Cardiac	8 (28%)	-	
Gastrointestinal	7 (24%)	-	
Renal	1 (3%)	-	
Current active disease	22 (76%)	-	
Eosinophilia (>500 cell/mm^3^)	10 (35%)	-	
**Ongoing pharmacological treatments**			
Systemic glucocorticoid	28 (96%)	-	
DMARDs	22 (76%)	-	
Mycophenolate mofetil	5 (17%)	-	
Cyclosporine	3 (10%)	-	
Azathioprine	2 (7%)	-	
Cyclophosphamide	1 (3%)	-	
Rituximab	1 (3%)	-	
Intravenous Immunoglobulin	2 (7%)	-	

## Data Availability

The 16S rRNA sequence data have been deposited in the NCBI Gene Expression Omnibus (GEO) repository (https://www.ncbi.nlm.nih.gov/geo/query/acc.cgi?acc=GSE203432, accessed on 20 March 2022).

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
