# Peer review of "Gut Microbiota and Associated Mucosal Immune Response in Eosinophilic Granulomatosis with Polyangiitis (EGPA)"

_biomedicines, 2022, doi:10.3390/biomedicines10061227_

Round 1
Reviewer 1 Report
This study demonstrate that there is not a substantial intestinal taxonomic dysbiosis in EGPA patients as compared to controls and an enrichment of potential pathobionts sustaining inflammation was observed in EGPA, particularly in patients with active disease. The analysis of the intestinal immune response revealed that a Th1/Th17-mediated response.
These results are new and generally, the manuscript is well written and the data are clearly presented.
I have the following comment that need to be addressed:
Concerning the analysis on the mucosal immune response, a comparison with healthy control or other disease is lacking. It will be important to discuss some published data on this aspect.
Author Response
First of all, we want to thank the reviewer for the favorable evaluation of our work and for the effort in revising our manuscript. Regarding its comment, we really thank him for raising this point, giving us the opportunity to improve the quality of our work. Sure, as we discussed, a limit of our study is the impossibility (for ethical reasons) of analyzing the mucosal immune response in enrolled healthy controls. Besides, as suggested by the reviewer, we enhanced our discussion by adding comparisons with other published data (see lines 352-364).
Reviewer 2 Report
I congratulate the authors to write a a very well structured manuscript. Except for some minor improvements, I suggest it to be accepted for publication.
Specific comments:
Line 219 and elsewhere: Please use italics for applicable genera or family names, if applicable
Line 220-221 and elsewhere: when using the term 'padj' the 'adj' part should be subscript. This would avoid the confusion for the reader
Author Response
We really appreciate the positive evaluation of the reviewer and his time dedicated to reviewing our manuscript. According to his suggestions, we applied the italics for genera names and used the subscript format for ‘adj’.